# Shared Genetic Architectures between Coronary Artery Disease and Type 2 Diabetes Mellitus in East Asian and European Populations

**DOI:** 10.3390/biomedicines12061243

**Published:** 2024-06-03

**Authors:** Xiaoyi Li, Zechen Zhou, Yujia Ma, Kexin Ding, Han Xiao, Dafang Chen, Na Liu

**Affiliations:** 1Department of Epidemiology and Biostatistics, School of Public Health, Peking University, Beijing 100191, China; sherryli@bjmu.edu.cn (X.L.); peevesomega@163.com (Z.Z.); 13260033986@163.com (Y.M.); 15801298733@163.com (K.D.); gesangmeiduo@pku.edu.cn (H.X.); 2Department of Neurology, Peking University Third Hospital, Beijing 100191, China

**Keywords:** coronary artery disease, type 2 diabetes mellitus, pleiotropy, multi-ancestry

## Abstract

Coronary artery disease (CAD) is a common comorbidity of type 2 diabetes mellitus (T2DM). However, the pathophysiology connecting these two phenotypes remains to be further understood. Combined analysis in multi-ethnic populations can help contribute to deepening our understanding of biological mechanisms caused by shared genetic loci. We applied genetic correlation analysis and then performed conditional and joint association analyses in Chinese, Japanese, and European populations to identify the genetic variants jointly associated with CAD and T2DM. Next, the associations between genes and the two traits were also explored. Finally, fine-mapping and functional enrichment analysis were employed to identify the potential causal variants and pathways. Genetic correlation results indicated significant genetic overlap between CAD and T2DM in the three populations. Over 10,000 shared signals were identified, and 587 were shared by East Asian and European populations. Fifty-six novel shared genes were found to have significant effects on both CAD and T2DM. Most loci were fine-mapped to plausible causal variant sets. Several similarities and differences of the involved genes in GO terms and KEGG pathways were revealed across East Asian and European populations. These findings highlight the importance of immunoregulation, neuroregulation, heart development, and the regulation of glucose metabolism in shared etiological mechanisms between CAD and T2DM.

## 1. Introduction

Coronary artery disease (CAD) is a global epidemic, and has steadily become the leading cause of global disease burden, in addition to neonatal disorders, in all age groups [1]. Type 2 diabetes mellitus (T2DM) is a common comorbidity of CAD, with the global prevalence of CAD in T2DM patients estimated at approximately 30% [2]. Previous evidence has shown that T2DM in non-pregnant women and men is an independent risk factor for CAD. In one study, patients with T2DM were found to have an approximately two-fold higher risk of developing and dying from CAD than controls, reflecting the cardiovascular toxicity caused by T2DM [3]. Several risk factors or clinical features of T2DM,including hyperinsulinemia, insulin resistance, and β-cell dysfunction, are associated with increased pulse wave velocity (PWV) and carotid intima-media thickness (CIMT), both non-invasive markers of CAD prediction [4,5]. Conversely, it has been reported that increased PWV and CIMT are associated with an increased risk of new-onset diabetes [6,7]. Obviously, the clinical relationship between CAD and T2DM has been well established. Given these connections, we hypothesized a shared genetic and molecular basis between CAD and T2DM. Several shared molecular mechanisms have been suggested, such as nuclear factor κB-mediated inflammation [8]. However, the shared mechanisms connecting the two phenotypes remain to be further elucidated. Understanding those shared pathogenic processes may have far-reaching utility for facilitating efforts to adjust treatment strategies that could delay or halt the progression of CAD and T2DM.

Shared pathogenic mechanisms between CAD and T2DM are influenced by genetic risk factors in addition to environmental factors. Heritability estimates vary around the median values of 72% for T2DM and 58% for CAD, showing that both traits are highly genetic [9,10]. Meanwhile, the polygenic nature of CAD and T2DM is becoming increasingly clear, as recent genome-wide association studies (GWASs) have identified many genetic loci associated with CAD and T2DM (237 loci for T2DM [11] and over 250 loci for CAD [12]). However, only a few variants were found to be associated with CAD and T2DM in single-trait GWASs at the genome-wide significance level, because these polygenic traits tend to have more causal and pleiotropic variants [single nucleotide polymorphisms (SNPs)] with small effect sizes [13]. Recent studies have suggested a positive genetic correlation between CAD and T2DM [14,15]. More needs to be known about the genetic architecture that CAD and T2DM share, and whether these shared associations represent a common signaling pathway. Leveraging pleiotropy, indicating a potential phenotypic effect for shared risk variants or genes, can help address the challenges in the common pathogenesis of both CAD and T2DM. Conditional and joint association analyses (association analysis based on subsets, ASSET) can identify the specific overlapping loci that are in either the same direction or opposite directions. These analyses allow some subsets of the studies to have no effect, identifying the strongest association signal and evaluating the significance of the variants while accounting for the multiple tests required by the subset search [16].

Of note, the majority of published GWASs have focused on people of European ancestry, thereby leading to a biased view of shared human genetics between CAD and T2DM because of differences in effect sizes, allele frequencies, and patterns of linkage disequilibrium (LD) across multi-ancestry populations, which may block the translation of shared genetic associations into clinical and public health practice for the global population [17]. Expanding GWAS analyses should be considered to systematically inspect the similarities and differences between shared genetic effects in different ethnic groups.

In this study, we applied genetic correlation estimations, bivariate genetic analyses at a single SNP level, and gene-level association analyses in East Asian (Chinese and Japanese) and European populations. Next, a series of bioinformatics analyses, including a fine-mapping model and functional enrichment method, was employed to identify the potential causal loci and explore the biological function of shared loci.

## 2. Materials and Methods

### 2.1. Individual Data for CAD and T2DM of Chinese Cohort

DNA samples and clinical data were collected from participants in a Beijing Fangshan cohort, which was designed to investigate atherosclerosis and related traits. Upon recruitment, participants were approximately 59 years of age (59.3  ±  8.7 years). CAD was defined as a history of myocardial infarction (MI), angina pectoris, ischemic cardiomyopathy, and self-reported diagnosis. A total of 605 CAD cases and 2042 controls were genotyped using a customized array in the current study. DM was diagnosed according to the following criteria: fasting plasma glucose ≥7.0 mmol/L, 2 h oral glucose tolerance test ≥11.1 mmol/L or a history of DM (self-reported and use of hypoglycemic drugs). Samples with self-report of Type 1 diabetes (T1DM) or insulin treatment within a year of diagnosis were excluded. This resulted in 680 T2DM cases and 2315 controls. The clinical characteristics of the CAD and T2DM cohorts have been described in the Appendix A. SNP genotyping for each sample was performed with the commercial release of the Illumina Infinium Human Exome-12 v1.2 BeadChip Array. Details regarding data collection, quality control, and the imputation and analysis of genotype data can be found in the Appendix A.

### 2.2. GWAS Summary Statistics for CAD and T2DM of European and Japanese Populations

GWAS summary statistics for CAD were obtained from the UK Biobank (UKBB) project, containing genotypic and phenotypic data collected on 408,458 Europeans [18,19]. All samples were diagnosed based on the PheWAS code 411, comprising 31,355 CAD cases and 377,103 controls [18,20]. For T2DM, GWAS summary statistics were sourced from the Diabetes Genetics Replication and Meta-Analysis (DIAGRAM) consortium [11]. The T2DM dataset consisted of 80,154 individuals with T2DM and 853,816 control individuals of European ancestry. In addition, we also collected GWAS summary data for CAD and T2DM from BioBank Japan (BBJ), a genome biobank that collaboratively collects DNA and phenotypes of samples from 12 medical institutions in Japan [21,22]. The cohort contained 45,383 T2DM cases and 132,032 controls, while the CAD cohort contained 29,319 cases and 183,134 controls. More details about sample selection, quality control, and association analysis can be found in previous publications in the literature [11,18,21,22]. All of the SNPs were lifted over to build 37 (Hg19) of the human reference genomes, and then strand-aligned with the 1000 Genomes release v3 (1000 G).

### 2.3. Global and Local Genetic Correlation Analysis

To qualitatively assess the polygenic overlap between CAD and T2DM, cross-trait linkage disequilibrium score regression (LDSR) was employed on the publicly available GWAS summary statistics of CAD and T2DM to estimate the SNP-based heritability of each trait and the global genetic correlation of CAD and T2DM [15,23]. A value of p < 0.05 indicated statistical significance. To ensure the repeatability of the result, we also used the GNOVA method, a more precise and powerful method for testing genetic correlation [24]. Furthermore, to qualitatively assess the shared genetic effects between the two phenotypes due to genetic variants at a locus level across East Asian and European populations (e.g., genome-wide partitioned LD blocks), we conducted local genetic correlation analyses with the local analysis of [co]variant annotation (LAVA) method [25]. Only the genetic regions identified by univariate analyses of each pair of traits were chosen for local bivariate genetic correlation analyses (p<0.05). The 1000G populations of Europe and East Asia were used as reference cohorts for defining the LD region of the corresponding populations. We used a Benjamini–Hochberg procedure to adjust the *p* value of genomic regions [26]. Detected regions were then annotated according to the proximity to annotated genes provided by gene position information from the Gencode database. 

### 2.4. Single-Variant Pleiotropic Analysis (ASSET)

ASSET was performed to identify the shared genetic variants between CAD and T2DM in datasets of different populations [16]. In fixed-effects meta-analyses, we set the genome-wide significance threshold of genetic variants as $P_{\mathrm{meta}}\;<\;5\;\times {\;10}^{-8}$, and the suggestive significance threshold as $5\;\times \;10^{-8}\;<{\;P}_{\mathrm{meta}}\;<\;5\;\times \;10^{-5}$. We also calculated the standardized genomic inflation factor to examine potential confounding effects [27]. After controlling for the standardized genomic inflation factor, the adjusted genomic inflation factor was near 1, suggesting no significant inflation of test statistics due to uncontrolled confounding effects. For every identified SNP, we determined whether significance was achieved using subset-based meta-analysis to explore the association directions and defined the true shared genetic effect of SNPs, including a one-sided test (variant was assumed to be associated with diseases in the same direction, *P_1side_*) and a two-sided test (variant was allowed to be associated with diseases in the opposite direction, *P_2side_*). For publicly available GWAS summary data of populations of European and East Asian (Japanese) ancestry, we repeated the ASSET analyses. The 1000 G population with corresponding ancestry was used as the reference panel for pairwise LD estimation.

### 2.5. Definition of Common Pleiotropic Loci and Functional Annotation

For common SNPs, their potential causal genes were defined. Genomic loci and corresponding lead SNPs were defined using the clump function of SNP2GENE of functional mapping and annotation (FUMA) protocol v1.5.2 (http://fuma.ctglab.nl/ (accessed on 5 April 2023)) [28]. SNPs with the most significant *p* value and independent of each other at LD $r^{2}\;<\;0.1$ were defined as the lead SNPs. Independent SNPs were defined as SNPs independent of each other at $r^{2}\;<\;0.6$ in a locus. A distinct genomic locus was defined as a genomic region containing genetic variants with distances less than or equal to 250 kb. We also performed functional annotation for all common variants. For more detail, see the Appendix A.

### 2.6. Multivariate Gene-Based Association Analysis

We mapped the significant noncommon variants identified by ASSET to genes using SNPnexus (https://www.snp-nexus.org/v4/ (accessed on 17 April 2023)) [29]. Multi-trait gene-based tests for noncommon variants (MATR) were performed to solve the problem of low statistical power of classical single variant-based association tests for low-frequency and rare variants. Following Luo et al., we annotated the variants with MAF < 5%, and the variants within the gene regions [annotated as exon, intron and noncoding RNA (ncRNA)] were selected for analysis [30]. If some noncommon SNPs were located in the genomic loci defined by FUMA, the genes in these loci were also considered in MATR.

### 2.7. Multiethnic Fine-Mapping of Shared Loci/Genes

Considering that the shared risk variants identified by ASSET may not have been truly pathogenic, we performed fine-mapping analyses with PAINTOR tools v3.1 to advance the prioritization of causal variants responsible for both CAD and T2DM [31]. We focused on shared loci/genes identified in Chinese populations and estimated the posterior probability (PP) of each SNP being causal. LD matrices were generated for European and East Asian ancestry groups separately with Plink2 using 1000 G reference panels, and the strand-ambiguous variants with alleles A/T or G/C were extracted. We further constructed 99% credible sets of causal variants (the smallest set of genetic variants that jointly have a PP of 99% containing at least one causal variant) by calculating the cumulative sum of PPs after descending until it was at least 99%. A lower number of variants in a credible set indicated a higher resolution of fine mapping.

### 2.8. Functional Enrichment Analysis in Datasets of Different Populations

To determine the biological functions or processes involving shared gene sets between CAD and T2DM, functional enrichment analyses (including Gene Ontology (GO) enrichment analysis and Kyoto Encyclopedia of Genes and Genomes (KEGG) pathway analysis) were run using the R package clusterProfiler v4.4, which is based on an overrepresentation approach [32,33]. In this analysis, the candidate genes from the result of functional annotation of loci using FUMA and those identified as shared genes for both traits in MTAR analysis were selected. The Benjamini–Hochberg method was used to adjust the *p* value of each GO term or KEGG pathway, with the cut-off of the adjusted *p* value set to 0.05.

A flow diagram of the overall study design is shown in Figure 1.

## 3. Results

### 3.1. Global and Local Genetic Correlations 

Global genetic correlations and *p* values from LDSR analyses including European and East Asian populations (Han Chinese and Japanese) are embedded in the top side of Figure 2a. No significant genetic correlation was observed for CAD and T2DM in the Chinese population, possibly because of the means $\chi^{2}\;(Mean\;\chi^{2}<1.02)$ of the summary statistics of both traits being too small. In contrast, CAD showed a moderate positive genetic correlation with T2DM ($r_{\mathrm{LDSR}}=0.38,\;P=3.86\;\times \;10^{-44}$) in the European population, and a smaller positive correlation in the Japanese population ($r_{\mathrm{LDSR}}=0.23,\;P=5.11\;\times \;10^{-9}$). For GNOVA, the results showed were similar to those from LDSC analysis, both in the Japanese ($r_{\mathrm{GNOVA}}=0.20,\;P=3.94\;\times \;10^{-25}$) and European populations ${(r}_{\mathrm{GNOVA}}=0.36,\;P=1.96\;\times \;10^{-113}$). 

Local genetic correlation analyses were performed (Figure 2b, Appendix A). We identified 45 genomic regions that showed significant local genetic correlation between CAD and T2DM (*P_BH-adj_* < 0.05) in the European population, and 18 genomic regions in the Japanese population. Next, we obtained the genes contained in genetic regions with a significant correlation and observed partial overlapping regions between the European population and East Asian population. Specifically, 25 genes on chromosome 6, including *HLA-DRA*, were overlapping between these two populations, while 8 genes on chromosome 16, such as *ATP5F1AP3*, exhibited overlap (Appendix A). However, the overwhelming majority of genetic regions with significant correlation differed substantially between the two populations, highlighting the marked heterogeneity of the genetic structure.

### 3.2. Shared Signals between CAD and T2DM in East Asian and European Populations

ASSET analyses were performed in East Asian and European populations to detect the potential shared SNPs between CAD and T2DM. Ninety-five SNPs achieved genome-wide suggestive significance in cross-trait meta-analyses (${5\times 10^{-8}<P}_{\mathrm{meta}}\leq 5\times 10^{-5}$) (Appendix A). To interpret these combined associations, we implemented subset-based analyses of suggestively significant SNPs and observed that 55 SNPs showed the same direction of effect for both CAD and T2DM because they were clustered into the same subgroups (Figure 3). Nineteen of the fifty-five shared signals were of low frequency (1%<minor allele frequency (MAF)≤5%) and rare frequency (MAF≤1%) (Appendix A), respectively. The top 10 significant variants were located in the chromosome 7p14.1 region, such as rs57400279 [$P_{\mathrm{meta}}=\;5.71\times 10^{-7},\mathrm{OR}=4.41(3.70–5.26$), *P_2side_* $=\;1.74\times 10^{-6},\mathrm{OR}=4.41(2.39–8.16)$]. For common SNPs (MAF ≥ 5%), rs10830045, located in chromosome 10q26.2 region, was most suggestively associated with CAD and T2DM [$P\;=\;1.27\times 10^{-5},\mathrm{OR}=1.59(1.55–1.63)$]. As for the Japanese population, 1707 shared signals were identified at the genome-wide significant significance level ($P_{\mathrm{meta}}\leq 5\times 10^{-8}$), and 6439 reached the suggestive significance level (Appendix A). The effects of all shared signals were in the same direction of effect for both traits, comprising 7888 common SNPs, 203 low-frequency SNPs, and 55 rare variants. The top 10 most significant SNPs were located in the chromosome 6p21.33 region and chromosome 9p34.3 region, such as the common SNPs rs2233965 [$P_{\mathrm{meta}}\;=\;3.14\times 10^{-17},\mathrm{OR}=0.95(0.95–0.95)$, *P_2side_* $=\;7.06\times 10^{-17},\mathrm{OR}=0.95(0.94–0.96)$] and rs10870136 [$P_{\mathrm{meta}}=\;7.56\times 10^{-16},\mathrm{OR}=0.94(0.94–0.94)$, *P_2side_* = 1.02 × 10^−15^, OR = 0.94(0.93–0.96)].

In the European population, a total of 28,729 signals, comprising 5783 at the significant significance and 20,858 at the suggestive significance level, were discovered to be associated with CAD and T2DM (Appendix A). All shared signals affected both traits in the same direction, including 27,183 common SNPS, 1308 low-frequency SNPS, and 238 rare variants. The top 10 most significant SNPs were located in chromosome 9p34.3 region, such as the common SNP rs7030641 [$P_{meta}\;=\;8.60\times 10^{-49},\mathrm{OR}=0.92(0.92–0.92)$, *P_2side_* = 8.13 × 10^−50^, OR = 0.92(0.91–0.93)].

### 3.3. Functional Annotation of Shared Common Loci

To identified the potential functions of shared common signals, we merged them into the genomic loci using FUMA. In the Chinese population, nine loci containing nine lead SNPs and ten additional independent SNPs were finally defined (Appendix A). In the Japanese and European populations, 166 (251 lead SNPs and 7633 additional independent SNPs) and 602 loci (931 lead SNPs and 26,252) were defined, respectively (Appendix A). The functional annotation of the candidate SNPs in these loci demonstrates that the majority were intronic and intergenic, no matter which population they were found in (Figure 4a,d,g). 1.8% of the SNPs (n =9) were exons that were distributed in 11 genes in the Chinese population, including *CHAC1* and *ADAT1* (Appendix A). Additionally, 3.4% (*n* = 3188) and 2.3% (*n* = 500) of the SNPs identified in the Japanese and European populations were exons.

12, 12,933, and 3234 candidate SNPs had combined annotation dependent depletion (CADD) scores above 12.37 in the Chinese, Japanese, and European populations, respectively, suggesting high deleteriousness (the results for the Chinese population are shown in Appendix A). Using the threshold of Regulome DB scores ≤ 2, 3.4% (*n* = 17), 5.7% (*n* = 961), and 2.6% (*n* = 1797) of candidate SNPs possessed strong regulatory potential (Figure 4b,e,h). The distribution of the minimum chromatin state shows that most of the candidate SNPs were located in the open regions of chromatin (Figure 4c,f,i). In the Chinese population, six lead SNPs and seven additional independent SNPs were rated 1 to 4, indicating that they were possibly engaged in the strong transcription of genes (Appendix A). A total of 111 SNPs and 2953 additional independent SNPs in the Japanese population, and 386 lead SNPs and 10,807 additional independent SNPs in the European population, were rated 1 to 4.

### 3.4. Gene-Based Association Analysis for Shared Noncommon Signals

We summarized the relationships of shared SNPs and corresponding mapped genes between populations (Figure 5). A total of 587 SNPs were shared between the Japanese and European populations, but there were no SNPs shared by the Chinese and European populations (Figure 5a). After mapping SNPs, 33, 776, and 2920 genes were identified as candidate genes in the Chinese, Japanese, and European populations, respectively, and were selected to employ in MTAR analyses. In addition, 300, 1, and 3 genes were shared by different pairs of populations (Figure 5b).

Centered at the noncommon variants, their mapped genes were considered in gene-based association analyses. At a significance threshold of $P\;<\;1.51\;\times \;10^{-3}$ (corresponding to 0.05/33), eight genes showed evidence for associations with both CAD and T2DM identified by at least one MTAR test (MTAR-o, cMTAR, or iMTAR). Compared with single-trait gene-based analysis (cctP), three genes, *DOCK1* ($P_{\mathrm{cMTAR}}=3.97\;\times \;10^{-4}$), *CHAC1* ($P_{\mathrm{cMTAR}}=1.28\;\times \;10^{-3}$), and *ZNF74* ($P_{\mathrm{cMTAR}}=3.78\;\times \;10^{-5}$), were exclusively identified by MTAR analyses (Table 1). The most significant gene was *TANGO2* ($P_{\mathrm{cMTAR}}=3.78\;\times \;10^{-5}$). Significant shared genes other than *MYOM2* have not been previously reported in genetic studies of CAD or T2DM.

In the Japanese population, the SNP sets of 42 genes were shown to be jointly associated with CAD and T2DM in at least one MTAR test ($P\;<\;6.44\;\times \;10^{-5}$, corresponding to 0.05/776), and 8 of 42 were exclusively found in MTAR analysis, such as *TAPBP* and *KIFC1* (Figure 5c, Appendix A). In addition, the 49 genes significantly associated with CAD and T2DM, identified by MTAR, have not been previously reported in previous studies. In the European population, 17 genes showed strong associations with both CAD and T2DM at $P\;<\;1.71\;\times \;10^{-5}(\mathrm{corresponding}\;\mathrm{to}\;0.05/2920)$, and 3 genes were exclusively significant, such as *TM4SF4*. 

### 3.5. Fine Mapping of Shared Loci/Genes between CAD and T2DM

For discovered shared loci/genes, we constructed 99% credible sets containing the causal SNPs that jointly accounted for 99% of the PP driving the association of the shared loci using GWAS summary statistics of the Chinese, Japanese, and European populations (Appendix A). At the specific ethnic level, all loci were fine-mapped to plausible causal variant sets less than or equal to 3, while 8 of the 9 were fine-mapped to plausible causal variant sets at multiple ethnic levels. In total, fine-mapping analysis finally prioritized 34 genetic variants as potentially causal for both CAD and T2DM, 34.0% (n=10) of which had comparatively high pp (≥99%); 5.9% of the variants (n=2) had a posterior probability between 90 and 99%. Interestingly, we observed that 4 loci had multiple SNPs with PP over 99%, suggesting a series of multiple causal SNPs located in these loci. For instance, the locus mapped to *CHAC1*, ranging from 41217043 to 41451357 on chromosome 6, had 2 putative casual variants (rs139348139 and rs75867673) with PPs of 98.28% and 99.25%.

### 3.6. Functional Enrichment Analysis of Shared Loci/Genes

Functional enrichment analysis of the candidate genes indicated the potential differences in related biological pathways in different populations for developing CAD and T2DM. Those candidate genes were significantly enriched in 171 GO terms ($P_{\mathrm{adj}}\;<\;0.05$), with the most enrichment results for the Japanese population dataset (*n* = 108) (Figure 6a and Appendix A). The most significant GO terms refer to immunoregulation, neuroregulation, and regulation of glucose metabolism, and 15 significant GO terms identified in the European population also showed significant enrichment in the Japanese population, referring to gene transcription and expression in the major histocompatibility complex (MHC) region related to the immune system. In the Chinese population, 1 GO term was significantly enriched, namely, the structural constituent of muscle (Appendix A), but most of the GO terms in which these candidate genes were involved are related to immunoregulation and heart development. For KEGG, we also identified 41 significantly enriched pathways in different populations, most of which were related to immunodeficiency disorder, abnormal metabolism of glucose/hormones, and infection (Figure 6b and Appendix A). Most of the candidate genes were also involved in these three kinds of pathways. Some differences were observed in different populations. For example, 1 gene identified in the Chinese population was assigned to the cytokine–cytokine receptor interaction pathway, whereas 27 in the European population and 4 in the Japanese population were associated with this same pathway, and they were not identical to each other. This reveals that the genes that play a pathogenic role in CAD and T2DM, although from the same pathway, were different across multiple populations.

## 4. Discussion

In this study, a significant global genetic correlation between CAD and T2DM was observed in both Japanese and European populations and the results of local genetic correlation provide supporting evidence for polygenic overlap among those traits in both populations. We then identified 55 suggestive shared signals in the Chinese population and over 20,000 signals in the Japanese and European populations. Functional annotation identified 300, 1, and 3 genes shared by the different populations. Gene-based analysis found 64 genes shared between CAD and T2DM. Functional enrichment analysis of candidate genes found that the candidate genes were significantly enriched in 171 GO terms and 41 KEGG pathways, the majority of which were associated with immunoregulation, neuroregulation, and the regulation of glucose metabolism. These findings support the importance of genetic structure in biological mechanisms shared by CAD and T2DM.

Several genes were successfully detected in both single SNP-level analysis and multivariate gene-based association analysis. For example, *CHAC1*, whose nearby SNPs [including rs10851400 (*P_2side_* $=\;3.50\times 10^{-5},\mathrm{OR}=0.25(1.24–1.26)$] were found to achieve suggestively significant levels in the Chinese population, encodes glutathione-specific gamma-glutamylcyclotransferase 1, which catalyzes the cleavage of glutathione into 5-oxo-L-proline and a Cys-Gly [34]. Additionally, *CHAC1* also achieved significance in the MTAR test ($P_{\mathrm{cMTAR}}=1.28\times \;10^{-3}$), suggesting the potential existence of a shared pathogenic mechanism between CAD and T2DM. *CHAC1* was assigned to a KEGG pathway known as glutathione metabolism in this study, indicating its critical role in the regulation of glutathione metabolism, which is involved in apoptosis and the oxidative balance of the cell, as descripted in a previous review [35]. Previous studies have discovered that increased *CHAC1* expression is correlated with glutathione depletion in human islets and the formation of atheromatous plaques [36,37]. A significant increase in cardiac mass and function at baseline was found in mice after genetic inhibition of *CHAC1*, suggesting that it may promote CAD development in humans [38]. Interestingly, a recent epigenome-wide DNA methylation study discovered low levels of DNA methylation within the *CHAC1* region in T2DM patients, and *CHAC1* knockdown by siRNA protected β-cells from apoptosis, suggesting that *CHAC1* is involved in apoptosis triggered after endoplasmic reticulum (ER) homeostasis in β-cells fails to re-establish in different branches of the ER stress response [38]. The direction of association we found is consistent with these findings. Thus, we presume that *CHAC1* may be a potential therapeutic target for the treatment of comorbid CAD and T2DM.

A missense mutation in the *ZNF74* gene [rs190749586, *P_2side_* $=\;2.59\times 10^{-5},\mathrm{OR}=2.77(2.46–3.12)$] was associated with both CAD and T2DM, and MTAR analysis validated it at the gene level ($P_{cMTAR}=3.78\;\times \;10^{-5}$). *ZNF74* is an important coding gene for the formation of RNA-binding protein tightly associated with the nuclear matrix, which has been reported to have a significant association with cardiac defects such as congenital heart disease [39]. *ZNF74*, targeted by miRNA let-7g, was correlated with cell cycle exit and cell differentiation in human pancreatic endocrine cells, indicating a non-neglectable effect of *ZNF74* on pancreatic islet function in T2DM [40]. This research suggested that the *ZNF74* gene may cause the occurrence of the two diseases by regulating the pathways of cell proliferation and differentiation. In addition, *MYOM2* was also defined as a pleiotropic gene of CAD and T2DM by MTAR (*P_MTAR-O_* $=3.57\times 10^{-6}$), with 11 novel pleiotropic SNPs discovered in the vicinity of this gene. Previous GWASs have found that other SNPs (rs755900673 and rs17064642) at this gene were associated with T2DM and adverse cardiovascular events, and the novel pleiotropic SNPs we identified provide more evidence for the shared mechanism between these two diseases [41,42]. *MYOM2* encodes a major component of the vertebrate myofibrillar M band, and hypomethylated CpG sites of *MYOM2* have previously been reported to be associated with atherosclerosis-related phenotypes [43]. 

In this study, 587 SNPs and 303 candidate genes were shared by East Asian and European populations. Those shared SNPs/genes reported in Chinese and Japanese populations successfully validated in European populations could be called trans-ancestry shared SNPs/genes. Based on GO and KEGG annotation or enrichment analysis, those shared genes were mainly enriched in the functions of immunoregulation, neuroregulation, and the regulation of glucose metabolism, reflecting extensive pleiotropy between CAD and T2DM. For example, the genes identified across the three populations, although not identical, were all assigned to neurodegeneration pathways, which is consistent with previous reports regarding CAD and T2DM [44,45]. However, among these shared genes, the majority only showed significant shared genetic effects in both the Japanese and European populations, while the minority showed significant genetic effects in both the Chinese and European populations, partly reflecting the ethnic heterogeneity of shared genetic structure patterns between the Chinese and other populations.

The pleiotropic loci between CAD and T2DM were validated by using genetic correlation analyses to a certain extent, which suggests that biological pleiotropy partly accounts for the high comorbidity of CAD and T2DM, and further captures the different patterns of shared genetic architecture among multiple populations. CAD showed a moderate positive global ${\;r}_{g}$ with T2DM in the European population and a smaller positive global ${\;r}_{g}$ in the Japanese population, in agreement with previous studies [14,46]. The potential reason for this may be that the discordant directions of local ${\;r}_{g}$ were identified in parts of significant genetic regions of the East Asian population compared to the results of the European population, concealing the true pleiotropic relationship between CAD and T2DM. It also suggests the interethnic genetic heterogeneity of the shared genetic architecture between two ancestries, which is consistent with the analysis results of ASSET, probably because of the mutually inverted genetic effects of colocalization of distinct ancestry-specific variants and the influence of natural selection [47]. 

This study has several strengths. Firstly, we identified 56 genes jointly associated with CAD and T2DM in the East Asian population and European population which have never been reported in previous studies, providing new clues for the treatment of comorbidities of CAD and T2DM. These discoveries offer potential therapeutic targets for developing novel drugs or gene therapy techniques, such as vascular endothelial growth factor gene therapy, which can accelerate the formation of new blood vessels in CAD and T1DM patients [48,49]. Additionally, they may help contribute to the development of predictive tools, such as polygenic scores, which can identify high-risk groups prone to CAD and T2DM comorbidity, and enable implementation of the precision treatment strategies. Secondly, to qualitatively elucidate the pleiotropic overlap between CAD and T2DM, the latest method of local genetic correlation, LAVA, was used, providing preliminary evidence for the exploration of shared genetic loci. The advantage of this tool is that it showed better control of type 1 error rates regardless of condition and a greater power to identify significant associations compared with the methods proposed previously, such as p-HESS and SUPERGNOVA [25]. Thirdly, the application of fine-mapping analysis and enrichment analysis not only partly validated our results, but also provided us with statistical evidence to verify the causal effect of pleiotropic loci of CAD and T2DM. These findings suggest a common genetic link between CAD and T2DM in terms of immune imbalance and abnormal metabolism of glucose/hormones.

Several limitations were also acknowledged. We were not well-equipped to detect shared genetic loci in the Chinese population because of the comparatively small sample size compared with the Japanese and European populations. In the European population and Japanese population, we detected more than 100 loci jointly associated with CAD and T2DM at a significant level, while we were able to detect only a few loci with suggestive significance in the Chinese population. More shared loci are expected to be discovered as sample input increases. 

## 5. Conclusions

In conclusion, the findings of this study highlight the importance of immunoregulation, neuroregulation, heart development, and the regulation of glucose metabolism in shared etiological mechanisms between CAD and T2DM, and reflect the differences across different populations. However, more molecular epidemiological studies with larger samples and experimental studies need to be conducted to validate the discovered pleiotropic genes.

## Figures and Tables

**Figure 1 biomedicines-12-01243-f001:**
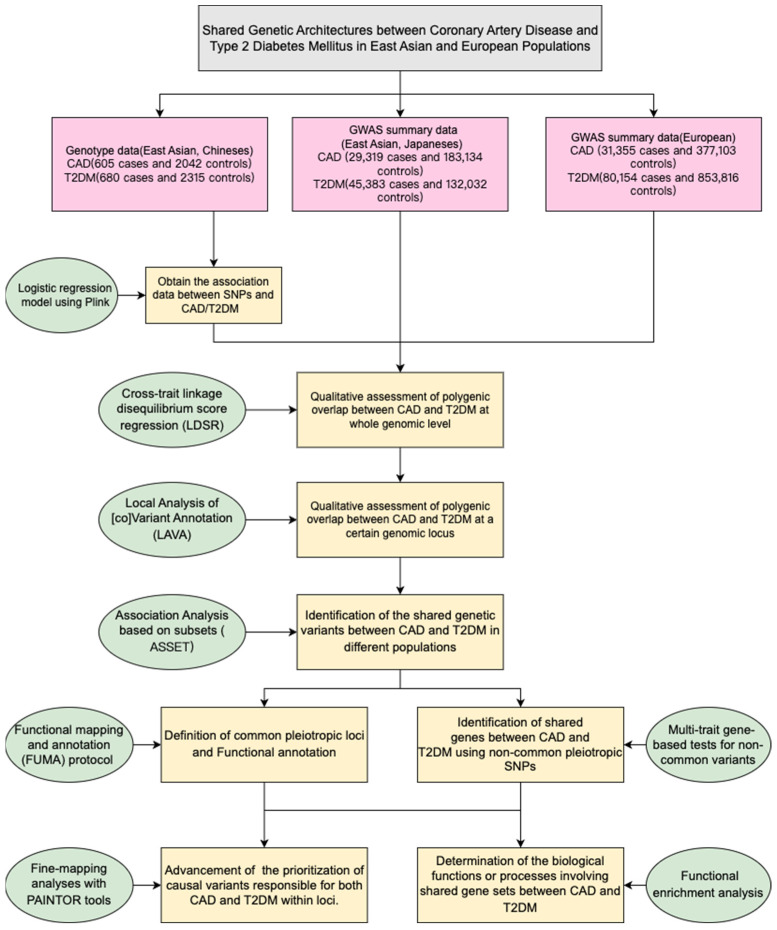
A flow diagram of the overall study design.

**Figure 2 biomedicines-12-01243-f002:**
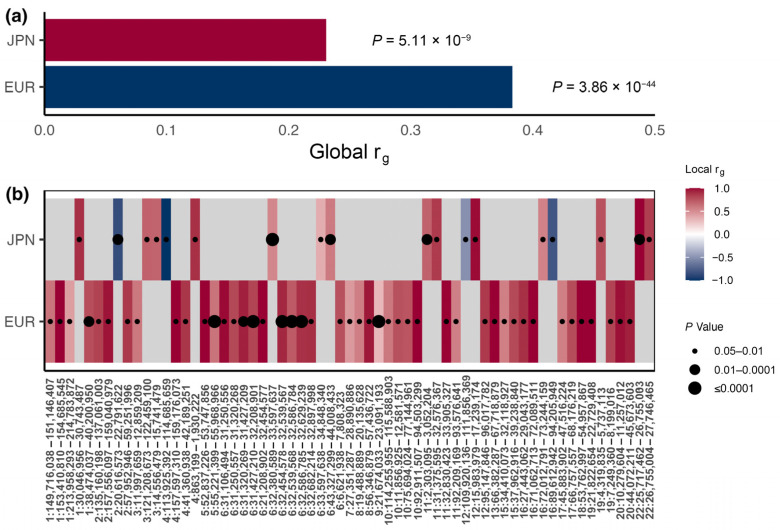
Global and local genetic correlations between CAD and T2DM in European (EUR) population and Japanese (JPN) populations. (**a**) represents the global correlation and *p* values (LDSR method). (**b**) represents the local correlation and Benjamini–Hochberg corrected *p* values (LAVA) in the European and Japanese populations in multiple genomic regions with significant correlation.

**Figure 3 biomedicines-12-01243-f003:**
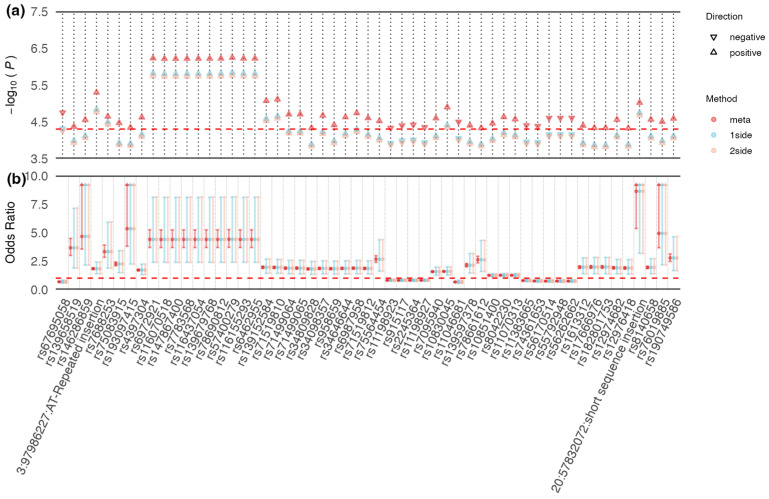
Discovery of 55 suggestively significant signals (${5\times 10^{-8}<P}_{\mathrm{meta}}\leq 5\times 10^{-5}$) in Chinese population. *p*-values and odds ratios (ORs) for each SNP in the meta-analysis results, one-sided subset test, and two-sided subset test are shown. (**a**) Negative log10-transformed *p* values are shown with respect to the shared SNP (rsID) and their chromosomal locations (Hg19). The direction of each triangle represents the direction of the shared genetic effect. The red horizontal dashed line indicates the negative log10-transformed *P* threshold $5\times 10^{-5}.$ (**b**) ORs and corresponding 95% confidence intervals are shown with respect to the shared SNPs and chromosomal locations. The red horizontal dashed line indicates OR = 1.0. Upwards arrows for some SNPs indicate that 97.5% of quantiles of confidence intervals are greater than or equal to 10.

**Figure 4 biomedicines-12-01243-f004:**
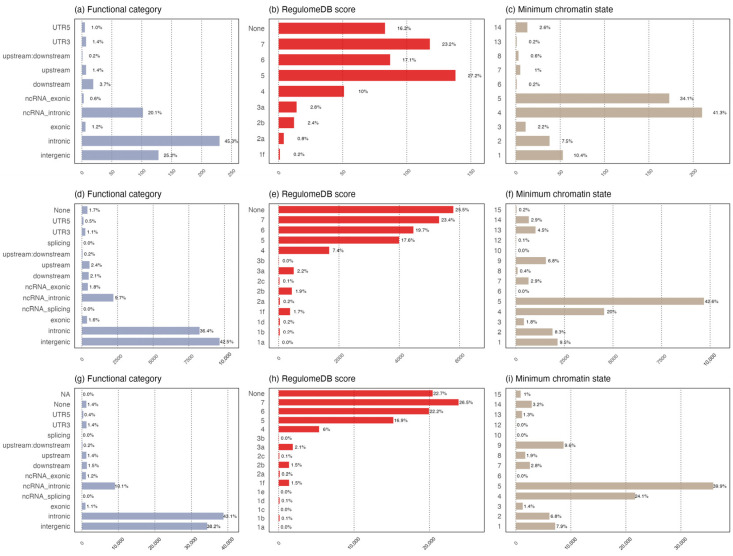
Bar charts of functional annotation for the candidate SNPs in common loci. Plots (**a**–**c**) represent the results of the Chinese population (9 common loci). Plots (**d**–**f**) represent the results of the Japanese population (166 common loci). Plots (**g**–**i**) represent the results of the European population (602 common loci). Plots (**a**,**d**,**g**) show the number of candidate SNPs with respect to the 12 kinds of functional consequences; plots (**b**,**e**,**h**) show the quantity of the distribution of candidate SNPs grouped by 7 levels of Regulome DB scores, where a lower score indicates a higher likelihood of having a regulatory function. Plots (**b**,**e**,**h**) show the quantity distribution of candidate SNPs grouped by 15 minimum chromatin states, with a lower state indicating higher accessibility of chromatin. States 1–7 refer to open chromatin states.

**Figure 5 biomedicines-12-01243-f005:**
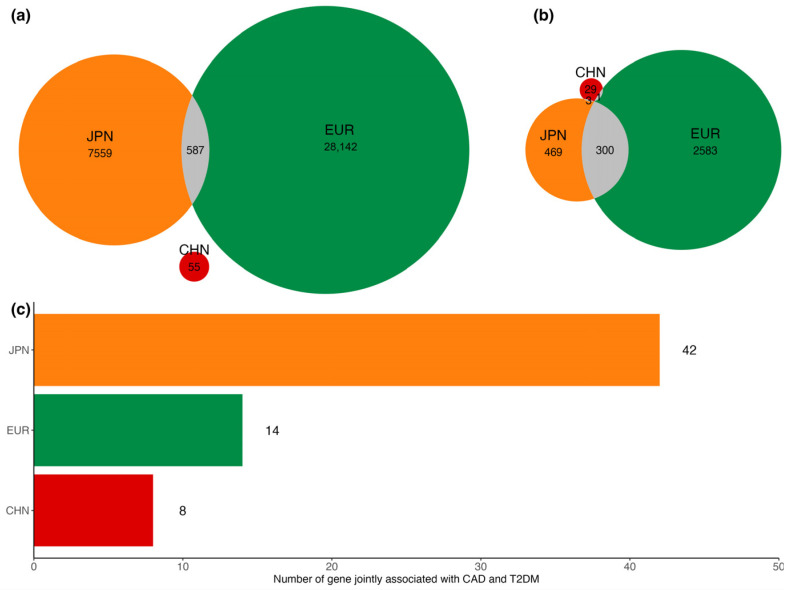
Shared SNPs, mapped genes, and significantly associated genes in Chinese, Japanese, and European populations. The Venn plots show the number of significant SNP level associations (ASSET) (**a**) and candidate genes identified by SNPnexus and FUMA (**b**). The bar graph (**c**) shows the number of significantly associated genes identified by MTAR. “CHN” represents Chinese population, “JPN” represents Japanese population, and “EUR” represents European population.

**Figure 6 biomedicines-12-01243-f006:**
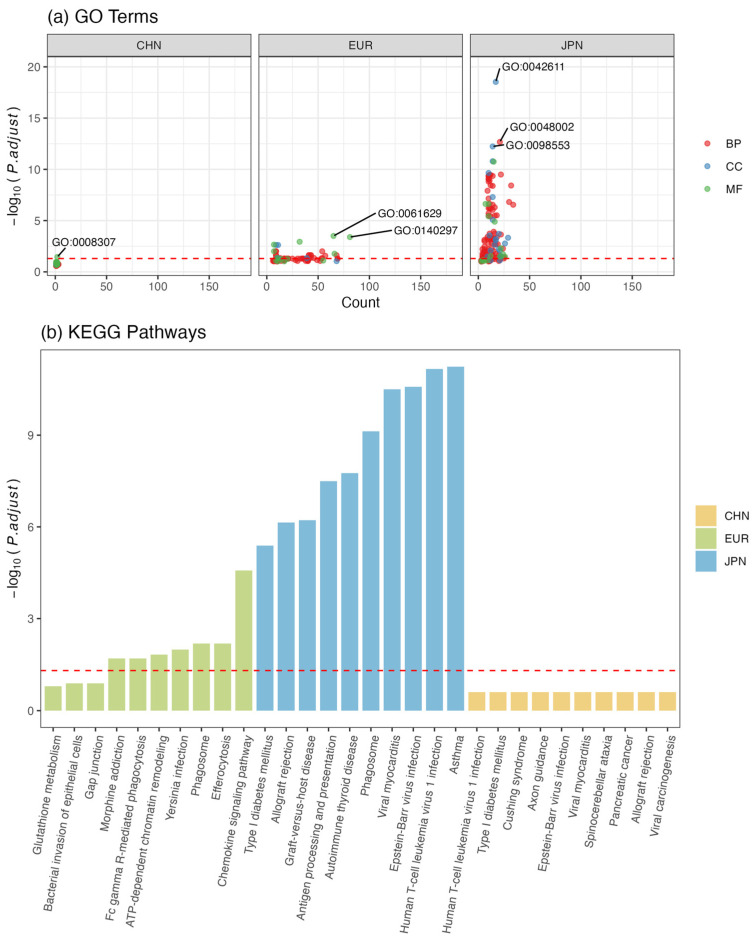
GO terms and KEGG pathways of candidate gene sets in different populations. (**a**) shows the negative log10-transformed *p* values of GO terms with respect to the number of enriched genes (count) in the aspect of biological processes (BP), cell component (CC) molecular functions (MF); (**b**) shows the top 10 enriched KEGG pathways. The horizontal axis represents the names of the KEGG pathways, while the longitudinal axis represents the negative log10-transformed *p* values of GO terms. The red horizontal dashed line indicates $P_{\mathrm{adj}}<0.05$.

**Table 1 biomedicines-12-01243-t001:** Results for eight genes detected by MTAR in Chinese population.

CHR	Gene	Number of SNPs ^1^	MTAR-O	cMTAR	iMTAR	cctP
4	*TBC1D9*	705	$5.66\times \;10^{-5}$	$5.01\times \;10^{-3}$	$4.48\times \;10^{-3}$	$1.90\times \;10^{-5}$
7	*AC072061.2*	269	$8.08\times \;10^{-6}$	$4.24\times \;10^{-6}$	$7.46\times \;10^{-6}$	$7.48\times \;10^{-4}$
8	*MYOM2*	1469	$3.57\times \;10^{-6}$	$1.97\times \;10^{-6}$	$3.02\times \;10^{-6}$	$7.80\times \;10^{-4}$
10	*DOCK1*	3541	$8.89\times \;10^{-4}$	$3.97\times \;10^{-4}$	$1.31\times \;10^{-3}$	$1.06\times \;10^{-2}$
15	*CHAC1*	20	$2.08\times \;10^{-3}$	$1.28\times \;10^{-3}$	$2.08\times \;10^{-3}$	$5.59\times \;10^{-3}$
16	*ADAT1*	166	$3.64\times \;10^{-4}$	$5.22\times \;10^{-4}$	$5.99\times \;10^{-4}$	$2.14\times \;10^{-4}$
22	*TANGO2*	375	$5.06\times \;10^{-7}$	$3.28\times \;10^{-7}$	$3.54\times \;10^{-7}$	$1.69\times \;10^{-5}$
22	*ZNF74*	6	$8.29\times \;10^{-5}$	$3.78\times \;10^{-5}$	$1.07\times \;10^{-4}$	$2.21\times \;10^{-3}$

^1^ The numbers of SNPs in the defined genomic loci. MTAR-O, cMTAR, iMTAR, and cctP are the *p* values of corresponding analysis method.

## Data Availability

For European, the data (GWAS summary statistics) for CAD and T2DM used in the analyses described here are freely available in the GWAS catalog website GCST008370 (https://www.ebi.ac.uk/gwas/studies/GCST008370, accessed on 3 September 2022) and DIAGRAM Consortium data download website (http://diagram-consortium.org/downloads.html, accessed on 3 September 2022), respectively. They were originally reported in Zhou, et al. and Mahajan, et al. [11,18]. For Japanese population, the data for CAD and T2DM were downloaded from the National Bioscience Database Center Human Database, with the accession code hum0014 (https://humandbs.dbcls.jp/en/hum0014-v32; accessed on 10 September 2022) and hum0197 (https://humandbs.dbcls.jp/en/hum0197-v3-220; accessed on 10 September 2022). They were originally reported in Ishigaki, et al. and Sakaue, et al. [21,22].

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
