# Peer review of "Shared Genetic Architectures between Coronary Artery Disease and Type 2 Diabetes Mellitus in East Asian and European Populations"

_biomedicines, 2024, doi:10.3390/biomedicines12061243_

Round 1
Reviewer 1 Report
Comments and Suggestions for Authors
Well written original paper with good design.
Abstract is clear and well written.
There is a lot of data, and at some point it is difficult to follow.
I would suggest to revise the discussion section, and to add some data of significant associations (like CHAC1, ZNF74..) and discuss potential mechanisms.
This will add to the importance of the paper
Author Response
We would like to express our sincere gratitude to you for the comments and suggestions on our article. We also submit a revised version of the manuscript and the corrections to our manuscript were all highlighted within the document by using red-colored text in this revised version.
Comments 1:There is a lot of data, and at some point it is difficult to follow.
Response: Thank you for your comments. I acknowledge the phenomenon you described does indeed exist. A large amount of genetic data was used for multiple analytical methods, resulting in numerous findings presented. In the original version of the manuscript, we presented the specific results that are significant to us, striving to offer readers a more tangible and intuitive understanding. Based on your feedback, we made the following modification:
(1) added descriptions of each column at the bottom of each supplementary table. For example, “Note” in table S1.
(2) the results in supplementary tables, described in main text, were highlighted in bold and placed at the top, ensuring they stand out prominently for readers.
(3) added more specific descriptions to certain results. For example, “Specifically,25 genes on chromosome 6, including HLA-DRA, were overlapping between these two populations, while 8 genes on chromosome 16, such as ATP5F1AP3, exhibited overlap.” (line 205-207).
(4) removed the most of the non-significant results from the enrichment analysis (table S9, tableS10)
Comments 2: Revise the discussion section, and to add some data of significant associations (like CHAC1, ZNF74.) and discuss potential mechanisms
Response: According to your suggestion, we added the message of significant associations while discussing some important genes. For example, “.... whose SNPs [including rs10851400] ……Additionally, CHAC1 also achieve significant level in MTAR test , suggesting the potential existence of shared pathogenic mechanism between CAD and T2DM.” (Line 369-408).

Reviewer 2 Report
Comments and Suggestions for Authors
In the present study, Li et al explored the genetic overlap between Chinese patients with CAD and diabetes and compared them with European and Japanase patients. They identified 56 new genes associated with both CAD and T2DM.
Overall the paper is interesting and uses innovative methodology. Moreover it brings new data to the current literature
I have only minor observations:
A more in-depth clinical description of Chinese patients is needed . To this purpose, authors should add a table in which clinical features of these patients are summarized. (CAD extension, previous myocardial infarction, comorbidities, etc) In patients with diabetes, the type of diabetes (1 or 2) should be specified In the discussion paragraph, line 313-314, authors state that “… , providing new clues for the treatment of comorbidities.”. Authors should explain how their genetic findings connect with the clinical management of patients with CAD and diabetes It is not clear to me why the method session has been postponed after the discussion. Please clarify this pointComments on the Quality of English Language
The paper needs only minor English language editing
Author Response
We would like to express our sincere gratitude to you for the comments and suggestions on our article. We also submit a revised version of the manuscript and the corrections to our manuscript were all highlighted within the document by using red-colored text in this revised version.
Comments 1:A more in-depth clinical description of Chinese patients is needed. To this purpose, authors should add a table in which clinical features of these patients are summarized. (CAD extension, previous myocardial infarction, comorbidities, etc)
Response: Thanks for your suggestion. To enable readers to better understand the specific information of the sample in the Chinese cohort, we have added 2 table and the coresponding description at the end of the Supplementary material to summarize the clinical characteristics. This was introduced by a sentence in the methods section of the manuscript: “The clinical characteristics of CAD and T2DM cohorts are descripted in the Supplementary material (Table S1 and Table S2).” (Line 92-94)
Comments 2: In patients with diabetes, the type of diabetes (1 or 2) should be specified.
Response: Thanks for your careful checks as we overlooked including the identification information for diabetes in this manuscript! We added a description of specify the diabetes type:” the samples with self-report of Type 1 diabetes(T1DM) or insulin treatment within a year of diagnosis were excluded.” (Line 90-92)
Comments 3: In the discussion paragraph, line 313-314, authors state that “… , providing new clues for the treatment of comorbidities.”. Authors should explain how their genetic findings connect with the clinical management of patients with CAD and diabetes.
Response: Thank you for your suggestions. We specific this sentence in 2 point.(1)offer potential therapeutic targets for developing novel drugs or gene therapy techniques;(2)it helps contribute to the development of predictor tools: “These discoveries offer potential therapeutic targets for developing novel drugs or gene therapy techniques, such as vascular endothelial growth factor gene therapy which can accelerate the formation of new blood vessels in CAD and T1DM patients[48, 49]. Additionally, they may help contribute to the development of predictive tools, such as polygenic scores, which can identify high-risk groups prone to CAD and T2DM comorbidity, and enable implementation of the precision treatment strategies.” (Line 438-443)
Comments 4: It is not clear to me why the method session has been postponed after the discussion. Please clarify this point.
Response: Thank you very much for reminding us. We found that we mixed up the order of the method session in the MDPI templates. Now it has been corrected.

Reviewer 3 Report
Comments and Suggestions for Authors
The present study is an interesting research into the genetic background of CAD and diabetes mellitus, revealing some common genetic loci. The results are all the more scientifically important as they derive from three different populations. The authors indicate also some specific mechanism (immunoregulation, regulation of glucose metabolism) coded by the candidate genes. One limitation on the data emerging from the chinese cohort could be its smaller sample as compares with European and Japanese populations. Some minor english language errors should be corrected (e.g.: line 86).
Nevertheless, this study has the potential to improve our knowledge of CAD mechanisms.
Author Response
We would like to express our sincere gratitude to you for the comments and suggestions on our article. We also submit a revised version of the manuscript and the corrections to our manuscript were all highlighted within the document by using red-colored text in this revised version.
Comments 1:One limitation on the data emerging from the chinese cohort could be its smaller sample as compares with European and Japanese populations.
Response: Thank you for your feedback. We acknowledge the disparity in sample sizes between the Chinese cohort and those of European and Japanese populations, thus we discussed it in the end of the manuscript. Additionally, we have devised a strategy to augment the sample size within the Chinese cohort for subsequent replication analyses over the coming years.
Comments 2:Some minor english language errors should be corrected.
Response: We have checked the manuscript carefully for grammar and spelling mistakes and made meticulous modifications to this manuscript. The corrected details are listed as highlighted by using red-colored text in the revised version.

Reviewer 4 Report
Comments and Suggestions for Authors
Li et al performed and analysis evaluating genetic Chinese, Japanese and European populations to identify the genetic variants jointly associated with CAD and type 2 diabetes. The results indicated significant genetic overlap between CAD and type 2 diabetes in those populations. Over 10,000 shared signals were identified and 587 were shared by East Asian and European populations. I find the study interesting, however, I think the results may be presented in a clearer way.
1. It is hard to understand and follow from the text, the number of cases from the Chinese, Japansese and European populaton.Could you add a flowchard or other type of scheme summarizing the proportion of each population and the different analysis it was included in?
2. Could you add more patient charasteristics? Currently, I see only mean age of the included cases. If you could add at least the proportion of male sex, this would be of interest to the readership.
Author Response
We would like to express our sincere gratitude to you for the comments and suggestions on our article. We also submit a revised version of the manuscript and the corrections to our manuscript were all highlighted within the document by using red-colored text in this revised version.
Comment 1 however, I think the results may be presented in a clearer way.
Response: Thank you for your feedback. I acknowledge the utilization of a significant amount of genetic data for multiple analytical methods, leading to numerous findings that may appear unclear. We have made the following modifications to address this issue:
(1) added descriptions of each column at the bottom of each supplementary table. For example, “Note” in table S1.
(2) the results in supplementary tables, described in main text, were highlighted in bold and placed at the top, ensuring they stand out prominently for readers.
(3) Provided more specific descriptions for certain results. For example, “Specifically,25 genes on chromosome 6, including HLA-DRA, were overlapping between these two populations, while 8 genes on chromosome 16, such as ATP5F1AP3, exhibited overlap.” (line 205-207).
(4) removed most of the non-significant results from the enrichment analysis. (Table S9, tableS10)
Comment 2 It is hard to understand and follow from the text, the number of cases from the Chinese, Japansese and European populaton.Could you add a flowchard or other type of scheme summarizing the proportion of each population and the different analysis it was included in?
Response: According to reviewer’s comment, we have added a flow diagram (Figure.1) at the end of method section to summarize our study design details, which was introduced by a sentence in the manuscript: “A flow diagram of overall study design is shown in Figure 1.”. (Line 186)
Comment 3 Could you add more patient charasteristics? Currently, I see only mean age of the included cases. If you could add at least the proportion of male sex, this would be of interest to the readership.
Response: Thanks for your suggestion. To enable readers to better understand the specific information of the sample in the Chinese cohort, we have added a table and a description at the end of the Supplementary material to summarize the clinical characteristics. This was introduced by a sentence in the methods section of the manuscript: “The clinical characteristics of CAD and T2DM cohorts are descripted in the Supplementary material (Table S1 and Table S2).” (Line 92-94)

Round 2
Reviewer 4 Report
Comments and Suggestions for Authors
The authors have adequatelly addressed my comments and the manuscript has improved substiantially.